# Identification of a master transcription factor and a regulatory mechanism for desiccation tolerance in the anhydrobiotic cell line Pv11

Takahiro G. Yamada[1], Yusuke Hiki[1], Noriko F. Hiroi[2], Elena Shagimardanova[3], Oleg Gusev[3,4], Richard Cornette[5], Takahiro Kikawada[5,6]*, Akira Funahashi[1]*

**1** Department of Biosciences and Informatics, Keio University, Yokohama, Kanagawa, Japan, **2** Faculty of Pharmaceutical Science, Sanyo-Onoda City University, Sanyo-Onoda, Yamaguchi, Japan, **3** Kazan Federal University, Kazan, Russia, **4** RIKEN Cluster for Science, Technology and Innovation Hub, RIKEN, Yokohama, Kanagawa, Japan, **5** Institute of Agrobiological Sciences, National Agriculture and Food Research Organization (NARO), Tsukuba, Ibaraki, Japan, **6** Graduate School of Frontier Sciences, The University of Tokyo, Kashiwa, Chiba, Japan

* kikawada@affrc.go.jp (TK); funa@bio.keio.ac.jp (AF)

**Data Availability Statement:** Almost all of relevant data are within the manuscript and its Supporting Information files. The RNA-seq datasets extracted,

## Abstract

Water is essential for living organisms. Terrestrial organisms are incessantly exposed to the stress of losing water, desiccation stress. Avoiding the mortality caused by desiccation stress, many organisms acquired molecular mechanisms to tolerate desiccation. Larvae of the African midge, *Polypedilum vanderplanki*, and its embryonic cell line Pv11 tolerate desiccation stress by entering an ametabolic state, anhydrobiosis, and return to active life after rehydration. The genes related to desiccation tolerance have been comprehensively analyzed, but transcriptional regulatory mechanisms to induce these genes after desiccation or rehydration remain unclear. Here, we comprehensively analyzed the gene regulatory network in Pv11 cells and compared it with that of *Drosophila melanogaster*, a desiccation sensitive species. We demonstrated that nuclear transcription factor Y subunit gamma-like, which is important for drought stress tolerance in plants, and its transcriptional regulation of downstream positive feedback loops have a pivotal role in regulating various anhydrobiosis-related genes. This study provides an initial insight into the systemic mechanism of desiccation tolerance.

## Introduction

Living organisms require water for active life. Losing water, desiccation stress, is lethal for most organisms. Terrestrial organisms are particularly vulnerable to desiccation stress, and have evolved mechanisms to tolerate it. Entering an ametabolic state (anhydrobiosis) enables organisms to survive even if more than 99% of body water is lost, to recover after rehydration, and to resume their normal life cycle [1]. Rotifers, tardigrades, nematodes, plants, and larvae of the African midge *Polypedilum vanderplanki* acquired the molecular mechanisms underlying anhydrobiosis and can avoid mortality caused by desiccation [2–6]. An embryonic cell line

used, and analyzed during the current study are deposited to DDBJ (accession number DRA008948).

**Funding:** Takahiro Kikawada received JSPS KAKENHI (Grant Number: 22128001, https://kaken. nii.ac.jp/en/grant/KAKENHI-ORGANIZER-22128001/), the European Union Horizon 2020 Research and Innovation Program, MSCA-RISE "DRYNET" (Grant Number 734434, https://cordis. europa.eu/project/id/734434), and a pilot program of international collaborative research (Collaborative research based on a joint call with Russia) under "Commissioned projects for promotion of strategic international collaborative research" (https://www.affrc.maff.go.jp/ kokusaikenkyu/attach/pdf/joint_call_for_ international_research-28.pdf) for sequencing of RNA-seq datasets used in this study. Elena Shagimardanova received Russian Science Foundation Joint Research Groups Grant 17-44-07002(https://rscf.ru/en/contests/search-projects/ 17-44-07002/) for preparation of RNA-seq libraries used in this study. Oleg Gusev received JSPS KAKENHI (Grant Number: 18H02217, https:// kaken.nii.ac.jp/en/grant/KAKENHI-PROJECT-18H02217/) for preparation of RNA-seq libraries used in this study. Takahiro Kikawada and Akira Funahashi received JSPS KAKENHI (Grant Number: 17H01511, https://kaken.nii.ac.jp/en/ grant/KAKENHI-PROJECT-17H01511/) for computational resources for all computational analyses in this study, i.e., DEG analysis, network inference, etc. and preparation of the manuscript, i.e., English proofreading.

**Competing interests:** The authors have declared that no competing interests exist.

established from *P. vanderplanki*, Pv11, is desiccation tolerant and is a tool to study anhydrobiosis [7]. After pretreatment with culture medium containing 600 mM trehalose, Pv11 cells survive 12 days of desiccation (<10% relative humidity) and resume proliferation a few days after rehydration [8]. Thus, the mechanism of the desiccation tolerance of *P. vanderplanki* is at the cellular level.

After the onset of desiccation stress, *P. vanderplanki* begins to synthesize and accumulate trehalose, a nonreducing disaccharide $C_{12}H_{22}O_{11}$ [9]. Trehalose is a compatible solute; according to the water replacement hypothesis, it protects phospholipid membranes and intracellular macromolecules [10]. Initially, trehalose accumulation is crucial for preventing lethal damage caused by desiccation. Then, genes encoding antioxidant proteins, late embryogenesis abundant (LEA) proteins, and protein-L-isoaspartate (D-aspartate) O-methyltransferases (PIMTs), which reduce the harmful effects of desiccation, are upregulated [11–15]. Antioxidant proteins reduce the harmful effects of reactive oxygen species (ROS) generated during desiccation and rehydration. LEA and PIMT proteins prevent protein denaturation caused by desiccation. The above genes are also upregulated in Pv11 cells after pretreatment with a high concentration of trehalose; thus, trehalose could play a role in the initiation of the transcription of anhydrobiosis-related genes [16]. Genes for oxygen-binding hemoglobins and aquaporins, which controls osmotic pressure on the phospholipid membrane, are upregulated [11, 17]. The genes related to the ribosome complex are downregulated, reducing energy consumption for protein translation; in desiccated Pv11 cells, no significant changes in gene expression have been detected and the expression of each gene is steady [16]. The addition of water to dried Pv11 cells significantly upregulates the genes related to DNA repair, especially to homologous recombination and nucleotide excision repair [16, 18]. These genes are needed not only to repair DNA damage that occurred during desiccation despite the activity of anhydrobiosis-related protective genes, but also to help resume proliferation.

Although various genes related to desiccation tolerance have been detected and molecular mechanisms for the anhydrobiosis of *P. vanderplanki* have been inferred [9–18], the transcription factors and regulatory mechanisms that drive the expression of these genes have not been clarified. Mazin et al. revealed that heat shock factor (HSF) has a substantial role in regulation of these genes [19]; the HSF-binding element is enriched in the enhancers of anhydrobiosis-related genes. In Pv11 cells, HSF knock-down significantly decreases transcription of the genes for the antioxidant protein thioredoxin (TRX), a LEA protein, and trehalose-6-phosphate phosphatase, but not that of other anhydrobiosis-related genes [19]. This knock-down does not completely suppress cell proliferation after rehydration, and some cells are able to recover. Thus, HSF is required but not sufficient to enhance the desiccation tolerance system. These data illustrate that transcription factors control anhydrobiosis and provide evidence of both regulation- and gene structure-level evolution in anhydrobiotic species.

Using Pv11 cells as a model, we conducted a transcription factor–oriented genome-wide study to clarify the anhydrobiosis transcriptional regulatory network. We showed that nuclear transcription factor Y subunit gamma-like (NF-YC), which is important for drought stress tolerance in plants [20], regulates various anhydrobiosis-related genes. We revealed that many positive feedback loops, which are important for the ON/OFF switching of the system [21], were regulated by NF-YC and could initiate the transcription of various anhydrobiosis-related genes, such as LEA, TRX, aquaporins, hemoglobins, as well as genes related to trehalose synthesis and DNA repair. We also showed that HSF is involved in the positive feedback loop regulation. To the best of our knowledge, this is the first report of the clarification of a comprehensive gene regulatory network for desiccation tolerance; we suggest that the driving mechanism to induce the anhydrobiosis-related genes consists mainly of NF-YC and many downstream positive feedback transcriptional loops. Thus, our study confirmed the role of

changes in the gene regulatory network as part of changes leading to the establishment of anhydrobiosis in insects.

## Results

### Quantification of gene expression by RNA-seq and analysis of differentially expressed genes during trehalose pretreatment and rehydration

We first obtained samples of total RNA from Pv11 cells under control conditions (T0) and at different stages of anhydrobiosis: trehalose pretreatment for 12 h (T12), 24 h (T24), 36 h (T36), or 48 h (T48), subsequent desiccation for 10 days (R0), and rehydration of R0 cells for 3 h (R3), 12 h (R12), 24 h (R24), or 72 h (R72). Experiments were performed in biological triplicate. All extracted RNA reads were mapped to the scaffolds of the *P. vanderplanki* genome; the mapping rates were higher than 90% except for R3, R12, and R24 (S1 Table). Not only the nuclear genome but also the mitochondrial genome is transcribed in Pv11 cells [22]. Mapping the reads of all samples to both genomes (S1 Fig) revealed that the mapping rates to the mitochondrial genome were much higher in R3, R12, and R24. To quantify the expression of each gene in all samples, we counted the mapped reads in each gene region using the *P. vanderplanki* annotated genome and performed principal component analysis (PCA) (Fig 1). The variance was considerably lower within each sample than between samples. A smaller distance between two points in a PCA plot indicates a more similar pattern of overall expression between the corresponding samples. Based on this mathematical principle, this result indicated that the expression patterns were more similar between the biological replicates than between different samples. Therefore, the samples were adequate for the intended analysis.

We performed differentially expressed gene (DEG) analysis and detected 9,558 DEGs for trehalose pretreatment and 11,997 for rehydration (likelihood ratio test, adjusted $p$-value $<$ 0.05, Benjamini-Hochberg (BH) method, S1 Data). The number of DEGs annotated with GO:003700 (DNA-binding transcription factor activity) was 5 for trehalose pretreatment, 31 for rehydration, and 76 for both (Fig 2A). The number of other DEGs, which we considered as the target genes (such as LEA genes), was 1,249 for trehalose pretreatment, 3,662 for rehydration, and 8,228 for both (Fig 2B). The number of DEGs for both trehalose and rehydration treatments was much higher than that for each individual treatment.

### Inference of transcriptional regulatory networks

Based on time-series expression data on differentially expressed transcription factors, we inferred the transcriptional regulatory networks activated during trehalose pretreatment and rehydration (S2 and S3 Figs). We then integrated these networks including the transcription factors and estimated regulations in both treatments (Fig 3). Feed-forward loop (FFL) and feedback loop (FBL) transcriptional regulation among any three transcription factors controls the characteristic dynamics of gene expression [21, 23]. FFLs can be classified into the coherent type (the same sign of direct and indirect regulation) and incoherent type (opposite signs of direct and indirect regulation) [24], and FBLs can be classified into positive and negative [23] (Fig 4A). The estimated network contained 11 coherent FFLs, 5 incoherent FFLs, 2 positive FBLs, and 2 negative FBLs (Fig 4B). In the randomization test, only the number of coherent FFLs was significantly higher ($p$-value = 0.0177) than that in a random network with the same number of nodes and edges. A coherent FFL is able to cut a transient input but transmit a sustained signal to the output [24], and acts as a noise filter. The fact that the transcriptional regulatory network in Pv11 cells contained a significant number of coherent FFLs implies that this desiccation tolerance system might accurately regulate anhydrobiosis-related genes.

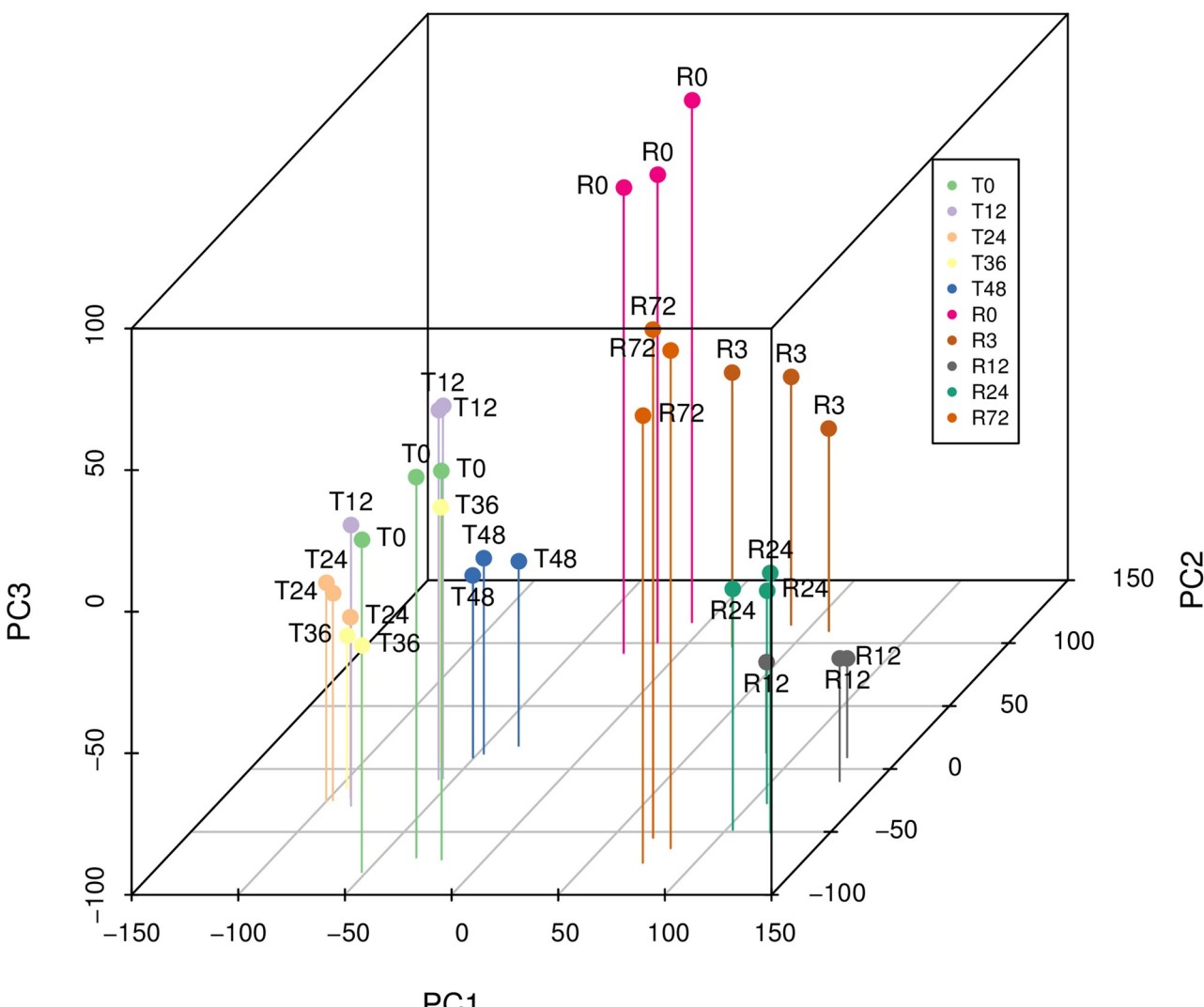

**Fig 1. Principal component analysis of all samples.** For each point on the plot, the label refers to the hours of trehalose pretreatment (T) or rehydration (R). The axes show the principal components (PCs). The contribution ratio was 23.6% for PC1, 14.4% for PC2, and 8.2% for PC3; the cumulative contribution ratio was 46.2%.

### Detection of groups of coexpressed genes (modules)

We detected 27 modules (Fig 5, S2 Data) and performed gene ontology (GO) enrichment analysis to identify the modules essential for desiccation tolerance (Fisher's exact test, $p$-value < 0.05, Table 1, S3 Data). Various GOs corresponding to gene functions reportedly necessary for desiccation tolerance were enriched in many modules (S1 Text). To annotate the modules containing anhydrobiosis-related genes, we counted the genes for LEA, TRX, PIMT, hemoglobin, aquaporin, and genes related to synthesis of trehalose and DNA repair in each module (S2 Table, S4 Data). Based on these results, we defined modules that included the maximum number of genes of each family previously reported as anhydrobiosis-related.

### Integration of transcriptional regulatory network and modules

We performed motif enrichment analysis to detect the transcription factor–binding elements enriched in the upstream regions of the genes in each module (CLOVER [25], randomization

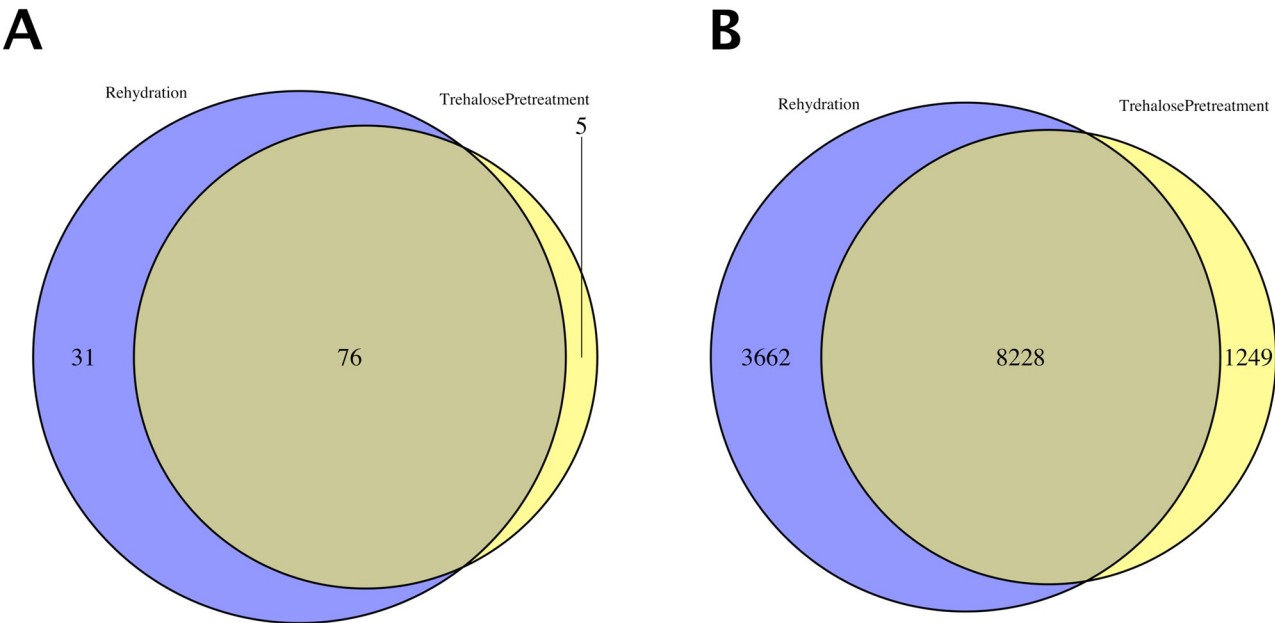

**Fig 2. Venn diagram of differentially expressed genes in trehalose pretreatment and rehydration.** (A) Transcription factor genes, (B) other genes (DESeq2, adjusted $p$-value < 0.05, Benjamini-Hochberg method).

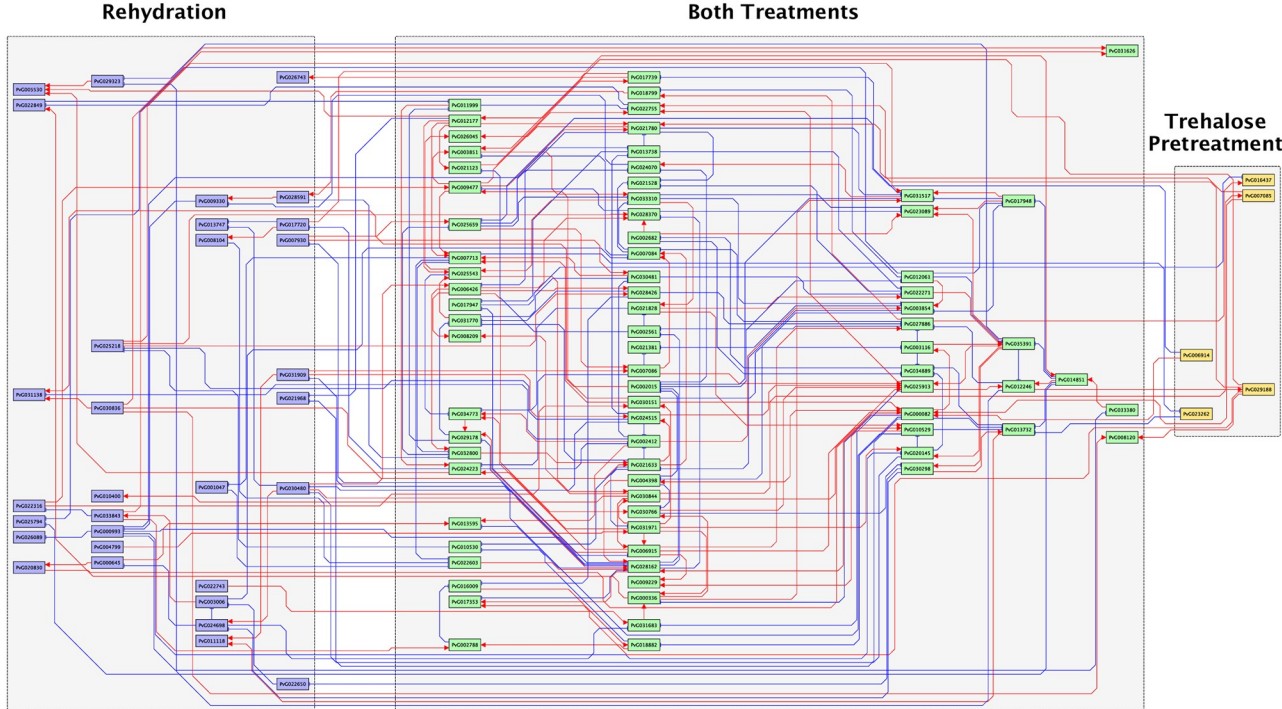

**Fig 3. Inferred transcriptional regulatory network obtained by integration of those of trehalose pretreatment and rehydration.** Each rectangular node refers to a transcription factor gene with the ID defined in the previous study [19]. Genes differentially expressed in trehalose pretreatment (yellow), in rehydration (blue), and both treatments (green) are shown. Red arrows, positive regulation; blue arrows, negative regulation.

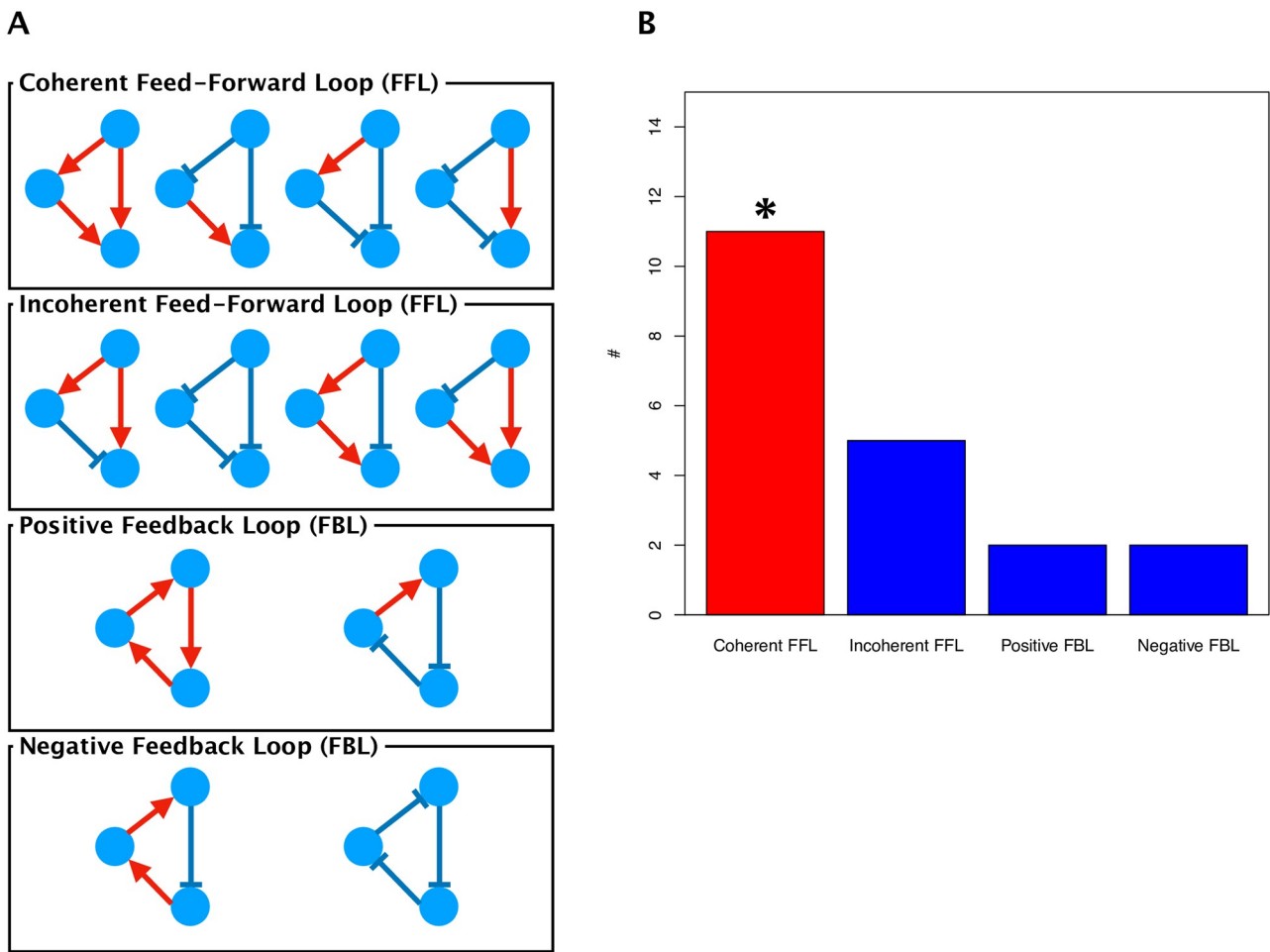

**Fig 4. Conceptual diagram of important structures in gene regulation in the integrated network.** (A) Structure of feed-forward loops (FFLs) and feedback loops (FBLs) [21], [23]. (B) The numbers of these loops in the integrated network. * Randomization test, $p$-value < 0.05.

test, $p$-value < 0.05; S5 Data). We also tested Granger causality between each transcription factor and the genes in the module at each time point (Granger causality test, adjusted $p$-value < 0.05, BH method; S5 Data). Finally, we extracted the regulatory relationships satisfying both criteria (Fig 6, Table 2, S5 Data). The results indicated that HSF (PvG018882) regulates all modules regardless of whether the regulation is direct or indirect. Thus, HSF was confirmed as a regulator of anhydrobiosis-related genes, consistent with the previous data [19].

## Comparative analysis of gene regulatory network between Pv11 cells and *Drosophila melanogaster*, a desiccation-sensitive species

We expected the estimated gene regulatory network to include a regulatory mechanism specific to desiccation tolerance in anhydrobiotes. We performed a comparative analysis between the estimated network and the transcriptional regulatory network of *D. melanogaster*, which is a desiccation-sensitive species [26] (S4 Fig), yet the results were too complex to understand which structure of gene regulation is crucial for desiccation tolerance. Therefore, we contracted the network, mainly by extracting the transcription factors that have no homologues in

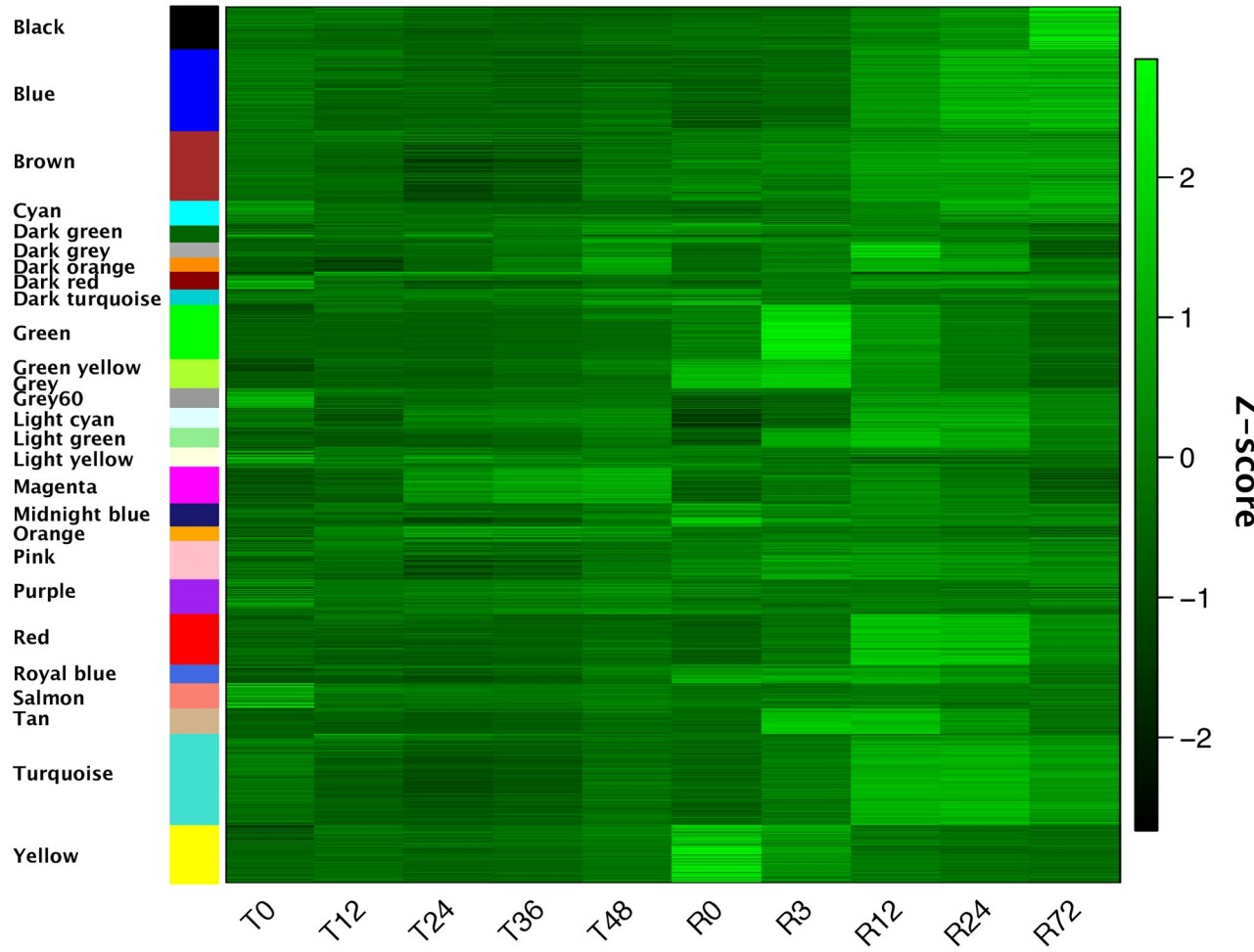

**Fig 5. Heatmap of the time-series Z-scores for all genes clustered in modules as detected by weighted gene coexpression network analysis.** Each module was assigned an easily recognizable color. Functional annotations (GOs) for each module are shown in Table 1, S2 Table and S3 Data.

the reported transcriptional regulatory network of *D. melanogaster* [26] and that directly regulated the modules (Fig 7). The contracted network was consistent with the previous reports [19] that HSF (PvG018882) is an important regulator of desiccation tolerance in *P. vanderplanki*. In the contracted network, PvG003116 was the most upstream transcription factor; it was homologous to nuclear transcription factor Y subunit gamma-like (NF-YC, accession number XP_029711307.1, blastp *e*-value = 1.0e-75). Most of the previously reported anhydrobiosis-related genes were regulated by PvG003116. The contracted network contained 3 FFLs and an FFL that included 3 FFLs, which we called TripleFFL (Fig 8). Four positive FBLs were downstream of NF-YC: between TripleFFL and FFL3; between FFL2 and FFL3; among TripleFFL, FFL2, and FFL3; and among HSF, FFL2, and TripleFFL. The modules that included the genes for LEA, TRX, aquaporin, hemoglobin, and the genes related to trehalose synthesis and DNA repair were regulated by NF-YC and these FBLs.

## Discussion

In our new RNA-seq datasets, the mapping rates of the R3, R12, and R24 samples were considerably lower for the nuclear genome and were much higher for the mitochondrial genome

**Table 1. Number of genes in each module and representative enriched GOs (GOs with the lowest *p*-value in Fisher's exact test).**

| Module | Number of Genes | Representative GO ID | GO Term | *p*-value |
|---|---|---|---|---|
| Black | 630 | GO:0046034 | ATP metabolic process | $8.69 \times 10^{-6}$ |
| Blue | 1222 | GO:0006260 | DNA replication | $1.20 \times 10^{-12}$ |
| Brown | 1045 | GO:0005488 | binding | $1.34 \times 10^{-13}$ |
| Cyan | 371 | GO:0016310 | phosphorylation | $2.96 \times 10^{-5}$ |
| Dark green | 255 | GO:0098800 | inner mitochondrial membrane protein complex | $1.11 \times 10^{-12}$ |
| Dark grey | 225 | GO:1901565 | organonitrogen compound catabolic process | $4.98 \times 10^{-4}$ |
| Dark orange | 213 | GO:0042254 | ribosome biogenesis | $3.62 \times 10^{-6}$ |
| Dark red | 265 | GO:0004719 | protein-L-isoaspartate (D-aspartate) O-methyltransferase activity | $8.38 \times 10^{-12}$ |
| Dark turquoise | 227 | GO:0009055 | electron transfer activity | $1.56 \times 10^{-4}$ |
| Green | 814 | GO:0008417 | fucosyltransferase activity | $2.17 \times 10^{-4}$ |
| Green yellow | 437 | GO:0042592 | homeostatic process | $5.11 \times 10^{-4}$ |
| Grey | 2 | No Hits | | |
| Grey60 | 296 | GO:0016667 | oxidoreductase activity, acting on a sulfur group of donors | $2.14 \times 10^{-4}$ |
| Light cyan | 298 | GO:0044444 | cytoplasmic part | $3.03 \times 10^{-8}$ |
| Light green | 294 | GO:0043227 | membrane-bounded organelle | $1.72 \times 10^{-8}$ |
| Light yellow | 289 | GO:0003735 | structural constituent of ribosome | $5.71 \times 10^{-98}$ |
| Magenta | 552 | GO:0140098 | catalytic activity, acting on RNA | $5.40 \times 10^{-10}$ |
| Midnight blue | 340 | GO:0016149 | translation release factor activity, codon specific | $5.04 \times 10^{-4}$ |
| Orange | 217 | GO:0008191 | metalloendopeptidase inhibitor activity | $9.04 \times 10^{-5}$ |
| Pink | 578 | GO:0005488 | binding | $1.35 \times 10^{-11}$ |
| Purple | 517 | GO:0004645 | phosphorylase activity | $1.71 \times 10^{-3}$ |
| Red | 760 | GO:0000502 | proteasome complex | $5.17 \times 10^{-24}$ |
| | | GO:1905369 | endopeptidase complex | $5.17 \times 10^{-24}$ |
| Royal blue | 279 | GO:0036459 | thiol-dependent ubiquitinyl hydrolase activity | $2.40 \times 10^{-4}$ |
| | | GO:0101005 | ubiquitinyl hydrolase activity | $2.40 \times 10^{-4}$ |
| | | GO:0019783 | ubiquitin-like protein-specific protease activity | $2.40 \times 10^{-4}$ |
| Salmon | 379 | GO:0000786 | nucleosome | $3.56 \times 10^{-13}$ |
| | | GO:0032993 | protein-DNA complex | $3.56 \times 10^{-13}$ |
| Tan | 381 | GO:0140097 | catalytic activity, acting on DNA | $9.47 \times 10^{-5}$ |
| Turquoise | 1366 | GO:0005488 | binding | $1.43 \times 10^{-9}$ |
| Yellow | 887 | GO:0044281 | small molecule metabolic process | $1.90 \times 10^{-4}$ |

than those of other samples (S1 Fig). These samples were obtained soon after the onset of rehydration after complete desiccation. Considering that mitochondria generate energy, we infer that the mechanism of recovery of Pv11 cells includes mitochondrial activation during rehydration. Currently, there are no reports of the active transcription of the mitochondrial genome just after rehydration. Subsequent detailed analysis of the process to activate the mitochondrial transcription may contribute to the discovery of the active recovery mechanism after rehydration.

In this study, the result of GO enrichment analysis for each module was not used for any prior knowledge to extract the finally contracted gene regulatory network (Fig 8). This result was used only for the verification of whether GOs related to the functions previously revealed as important for desiccation tolerance are enriched in modules or not (S1 Text). This result showed that almost all of the important GOs for desiccation tolerance was enriched in any of the modules. However, there are various GOs that were not focused on this verification

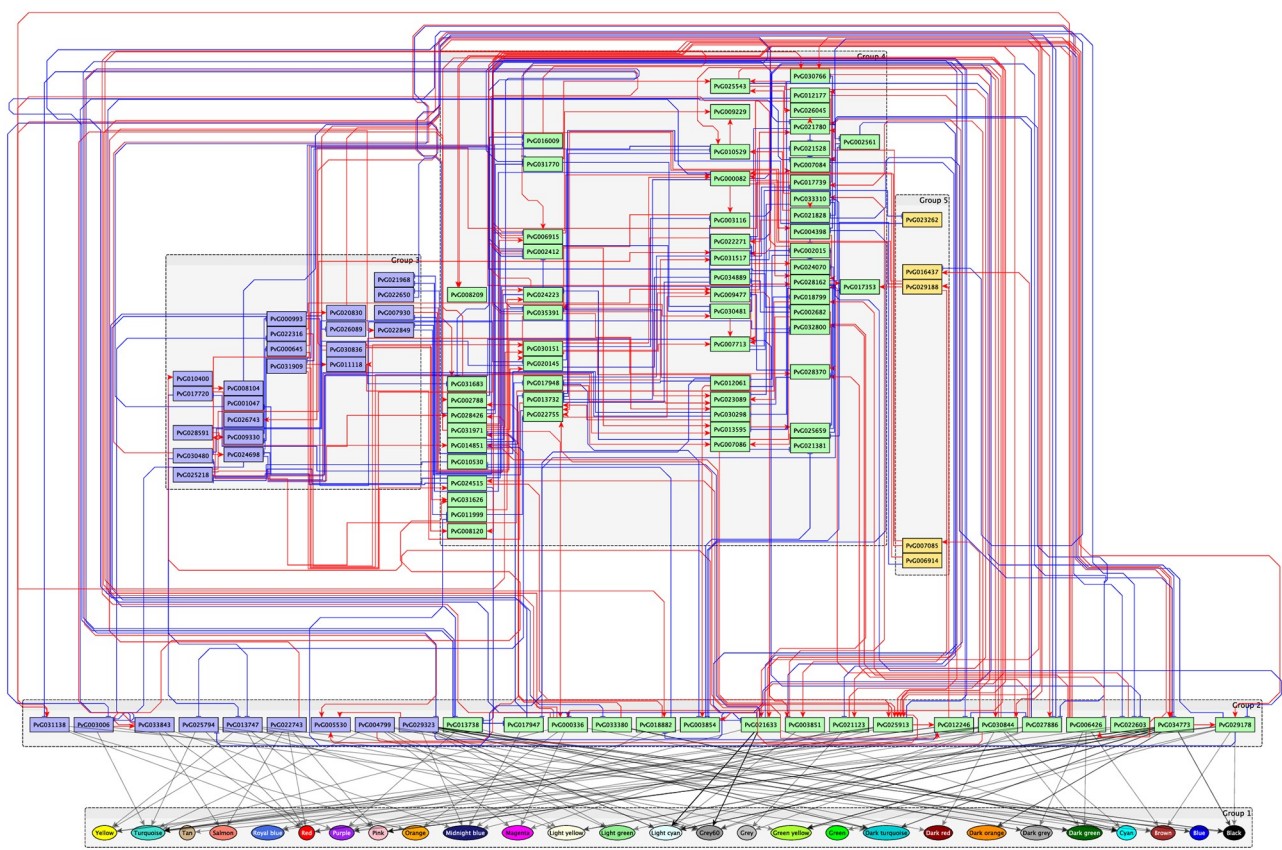

**Fig 6. Inferred gene regulatory network with the integration of the regulatory relationships of transcription factors and modules.** Rectangular nodes, transcription factors; circular nodes, modules; arrows, inferred regulatory relationships. The colors of rectangular nodes and arrows between transcription factors are the same as in Fig 3. The colors of the circular nodes are the same as in Fig 5. Regulatory relationships between transcription factors and modules are shown in grey (CLOVER, *p*-value < 0.05 and Granger causality test, adjusted *p*-value < 0.05, Benjamini-Hochberg method).

(S3 Data), and the relationships between these GOs and desiccation tolerance was not experimentally clarified yet. The subsequent detailed analysis of these GOs could lead to the discovery of novel molecular biological mechanisms of desiccation tolerance.

Our analysis indicates that NF-YC is the most upstream transcription factor regulating various anhydrobiosis-related genes. It is one of the three subunits of nuclear factor Y (NF-Y) [27], the other two being NF-YA and NF-YB. NF-YC binds to the CCAAT motif through the interaction with NF-YA and NF-YB and regulates the transcription of various genes [28]. Transgenic *Arabidopsis thaliana* constitutively expressing NF-YB was alive even after 8 days of severe drought stress, whereas wild-type plants were completely wilted [29]. The overexpression of NF-YC in rice enables it to stay turgid after 10 days of water withholding, whereas the wild-type rice showed extreme wilting [20]. These studies indicate that NF-Y is essential for drought stress tolerance in plants, but there have been no reports of the relationship between NF-Y and desiccation tolerance in animals. To our knowledge, this is the first report implicating NF-Y in the extreme desiccation tolerance of an anhydrobiotic animal. We examined NF-YC expression at the organismal level by re-analyzing our previously reported dataset [14]. The expression increased after desiccation and rehydration in *P. vanderplanki* larvae (24 h of desiccation/control ratio = 1.22; 3 h of rehydration/48 h of desiccation ratio = 7.22). *Polypedilum nubifer* is closely related to *P. vanderplanki* but is desiccation sensitive; it has an NF-YC

**Table 2. Direct regulation of modules by transcription factors and dependency on heat shock factor (HSF, PvG018882) for each module.**

| Module | Direct Regulators | HSF(PvG018882) Dependency |
|---|---|---|
| Black | PvG029178; PvG034773; PvG029323; PvG031138 | TRUE |
| Blue | PvG029323 | TRUE |
| Brown | PvG029178; PvG006426; PvG000336; PvG013747 | TRUE |
| Cyan | PvG030844; PvG021633; PvG033380; PvG004799; | TRUE |
| Dark green | PvG006426; PvG030844; PvG029323; PvG025794; PvG005038 | TRUE |
| Dark grey | PvG006426; PvG029323 | TRUE |
| Dark orange | PvG034773; PvG003458; PvG005038 | TRUE |
| Dark red | PvG030844; PvG004799; PvG005038 | TRUE |
| Dark turquoise | PvG013738; PvG034773; PvG003006; PvG029323 | TRUE |
| Green | PvG029178 | TRUE |
| Green yellow | PvG029178; PvG034773; PvG013747; PvG005038 | TRUE |
| Grey | PvG027886; PvG021123; | TRUE |
| Grey60 | PvG027886; PvG030844; PvG021633; PvG029323; PvG025794; PvG005530 | TRUE |
| Light cyan | PvG030844; PvG021633; PvG003851; PvG034773; PvG029323 | TRUE |
| Light green | PvG022603; PvG029323; PvG004799 | TRUE |
| Light yellow | PvG034773; PvG004799; PvG013747; PvG003458 | TRUE |
| Magenta | PvG021123; PvG000336; PvG004799 | TRUE |
| Midnight blue | PvG025913; PvG018882; PvG017947; PvG033843; PvG029323 | TRUE |
| Orange | PvG000336; PvG031138; | TRUE |
| Pink | PvG030844; PvG034773; PvG022743; PvG013747 | TRUE |
| Purple | PvG027886; PvG029178; PvG021633; PvG013738; PvG031138 | TRUE |
| Red | PvG022603; PvG030844; PvG033843; PvG022743; PvG013747; PvG025794; PvG005038 | TRUE |
| Royal blue | PvG025913; PvG003854; | TRUE |
| Salmon | PvG033843; PvG022743; PvG005038 | TRUE |
| Tan | PvG029178 | TRUE |
| Turquoise | PvG025913; PvG027886; PvG029178; PvG030844; PvG012246; PvG000336; PvG003006; PvG004799; PvG025794; PvG003458; PvG005038 | TRUE |
| Yellow | PvG025913; PvG030844; PvG022743; PvG013747; PvG005038 | TRUE |

homologue, Pn.05756 (blastn, *e*-value = 9e-32), and the increase in its expression was similar to that in *P. vanderplanki* (24 h of desiccation/control = 1.21) [14]. In *D. melanogaster*, nuclear factor Y-box C (FBtr0070974) is homologous to *P. vanderplanki* NF-YC (tblastn, *e*-value = 1.35e-38), but there are no reports about its expression under desiccation stress. We infer that the NF-YC gene is conserved in insects and contributes to mild desiccation tolerance; in *P. vanderplanki*, NF-YC has evolved to strengthen desiccation tolerance.

Why is *P. nubifer* not desiccation tolerant despite an increase in the expression of NF-YC similar to that of *P. vanderplanki*? Our estimated network could provide an answer to this question: the transcription factors connecting NF-YC to the modules that includes anhydrobiosis-related genes are absent in *P. nubifer*. These transcription factors include (1) PvG02163 in FFL2, which regulates FFL1 and modules containing genes for LEA, aquaporin, and hemoglobin, and trehalose-related genes; (2) PvG024515 in FFL2, which

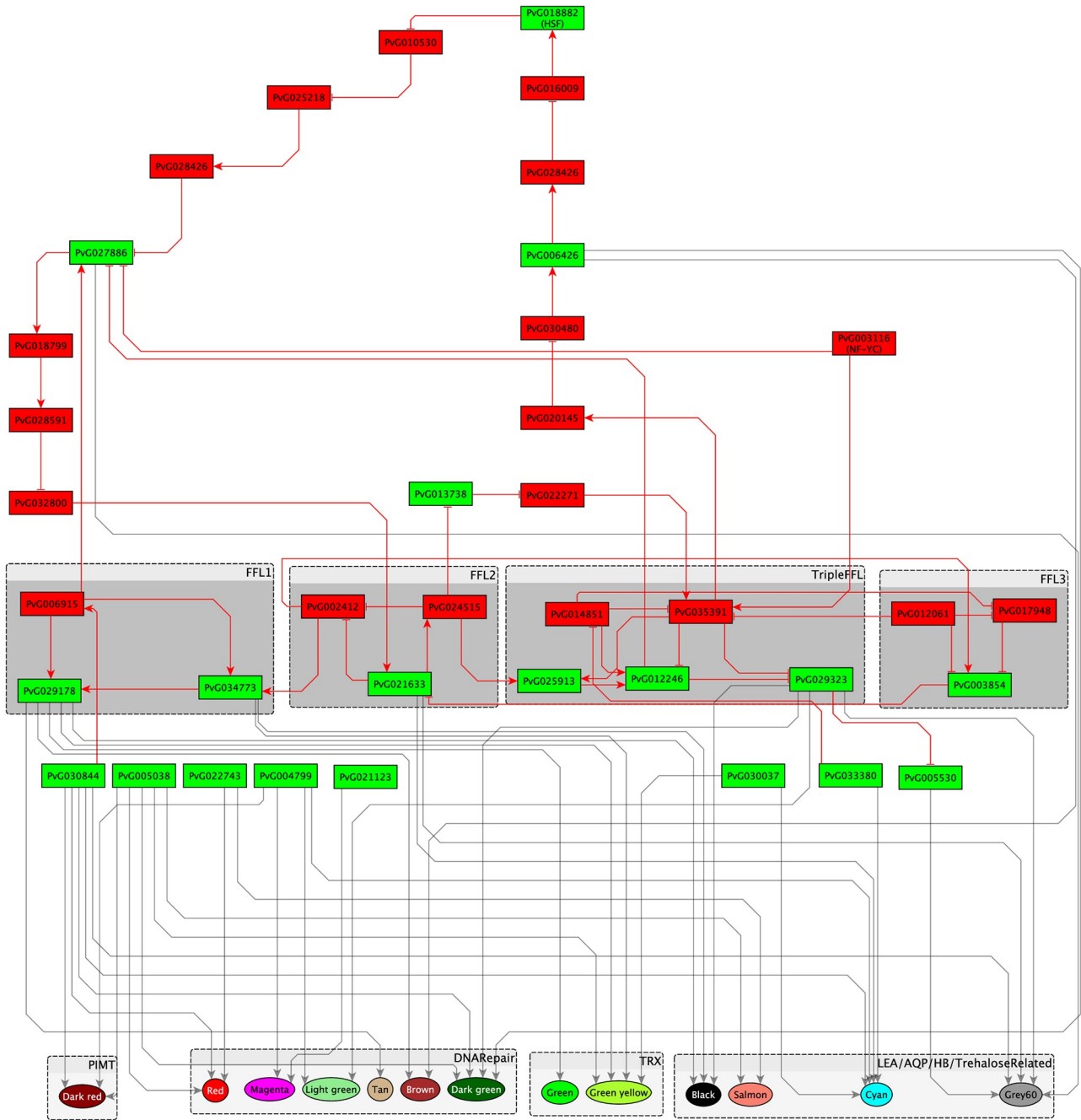

**Fig 7. Contracted network obtained from the comparative analysis of the transcriptional regulatory networks between *D. melanogaster* and Pv11 cells.** Rectangular nodes, transcription factors; circular nodes, modules; arrows, inferred regulatory relationships. Transcription factors with sequence similarity to the transcription factors in the *D. melanogaster* transcriptional regulatory network are shown in green (blastp, *e*-value < 1.0e-15); other transcription factors are shown in red. Regulatory relationships confirmed in both Pv11 cells and *D. melanogaster* are shown as green arrows, and those specific for Pv11 cells are shown as red arrows. Transcription factors that are components of feed-forward loops (FFLs) are shown in blocks FFL1, FFL2, FFL3, and TripleFFL.

regulates FFL1 and TRX and DNA repair modules; and (3) PvG029323 in TripleFFL (Fig 7), which regulates modules containing genes for LEA, aquaporin, and hemoglobin, and trehalose- and DNA repair–related modules (blastn, *e*-value < 1.0e-5) [14]. This result implies that *P. nubifer* is not desiccation tolerant because of the absence not only of anhydrobiosis-

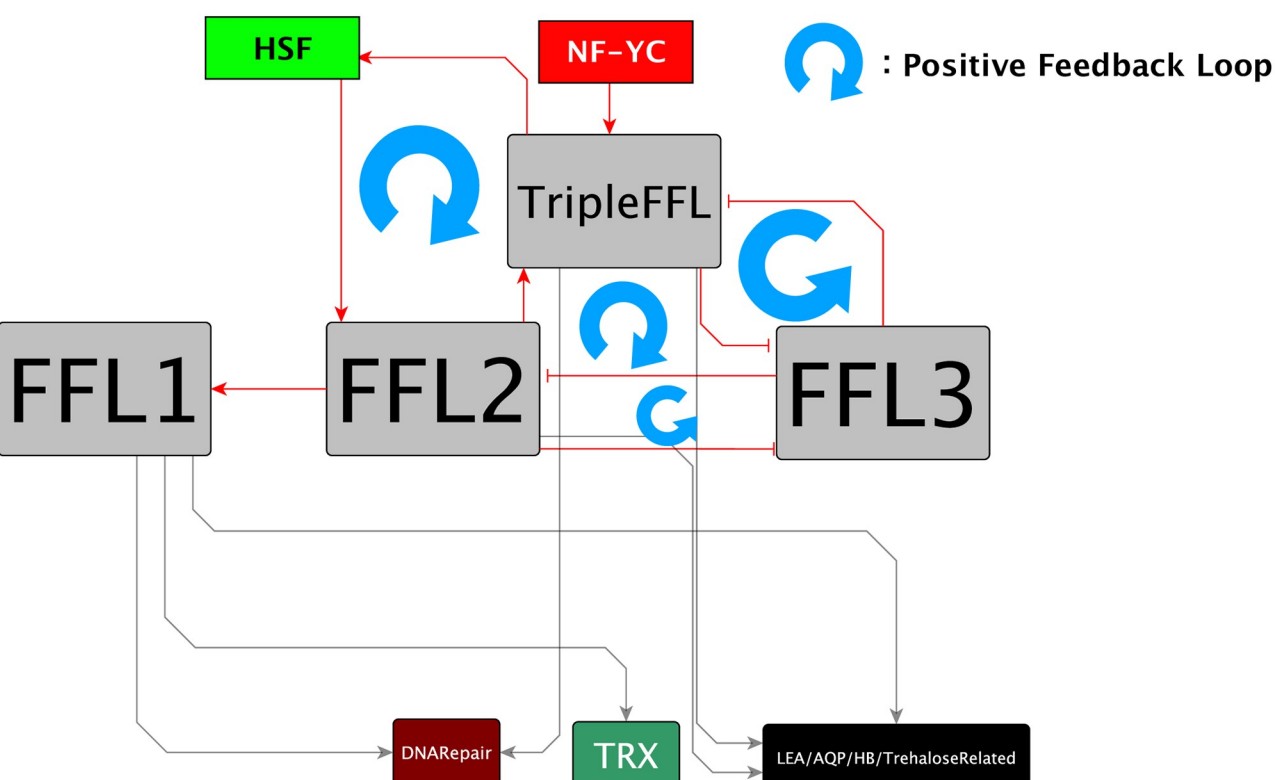

**Fig 8. Final contracted gene regulatory network for desiccation tolerance in Pv11 cells.** The network includes four positive FBLs: between TripleFFL and FFL3; between FFL2 and FFL3; among TripleFFL, FFL2, and FFL3; and among HSF, TripleFFL, and FFL2. Modules that include the anhydrobiosis-related genes for LEA, thioredoxin (TRX), hemoglobin (HB), aquaporin (AQP), and the genes related to the synthesis of trehalose and DNA repair are regulated by NF-YC homologue through the 4 positive FBLs.

related genes, such as LEA genes, but also of critical transcription factor genes regulating anhydrobiosis-related genes.

Our estimated gene regulatory network includes several positive FBLs from NF-YC to modules that include anhydrobiosis-related genes (Fig 8). A positive FBL of transcriptional regulation controls the activation (ON) or inactivation (OFF) of the system depending on the strength of the input signal [21]. Four positive FBLs in the estimated gene regulatory network imply that the transcription of anhydrobiosis-related genes is activated by trehalose pretreatment but not in the normal hydrated state (Fig 9).

Our estimated gene regulatory network is sufficient to explain various previous data on the expression of anhydrobiosis-related genes. However, the essential and required structure of gene regulation in the estimated network has not been tested yet. A recent study used RNAi-mediated knock-down in Pv11 cells and achieved silencing of a specific gene [30]. This gene silencing technique would be useful to test which transcription factor in our estimated network, such as NF-YC or a component of a positive FBL, is essential for desiccation tolerance; a significant decrease in the survival rate of Pv11 cells would be expected after gene silencing of such a transcription factor. By inferring *de novo* a gene regulatory network, we discovered the putative induction mechanism of desiccation tolerance. The estimated network also gave us the insight into the acquisition of desiccation tolerance in evolution and may allow us to control desiccation tolerance in the future.

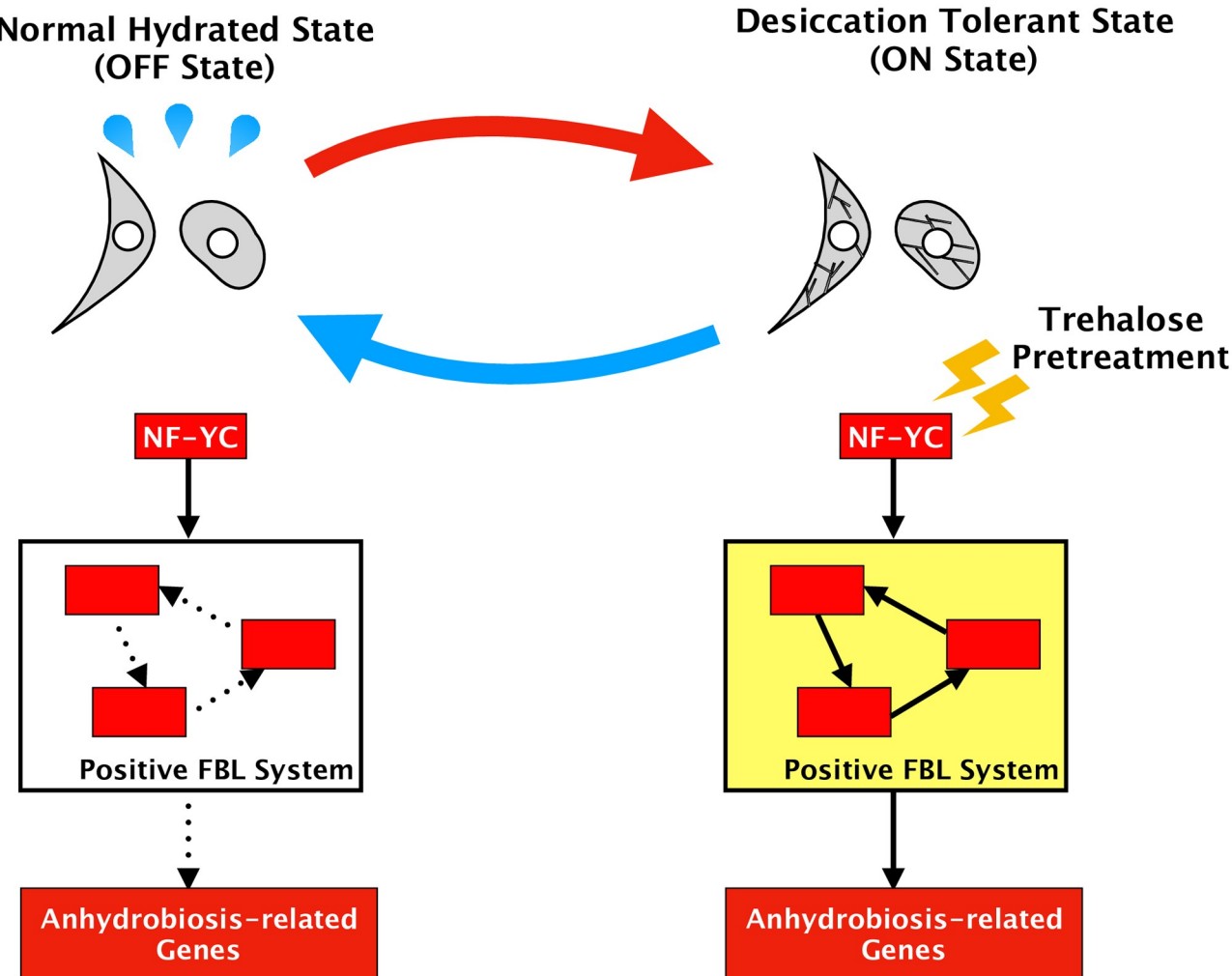

**Fig 9. Conceptual diagram of the inferred ON/OFF mechanism of desiccation tolerance in Pv11 cells.** The inferred gene regulatory network for desiccation tolerance contains four positive FBLs (Fig 8). A positive FBL of transcriptional regulation can either activate (ON) or inactivate (OFF) the system depending on the strength of the input signal [21]. This property and our results suggest that the activation or inactivation of anhydrobiosis-related genes may be controlled by these positive FBLs.

## Materials and methods

### RNA sample preparation, Illumina HiSeq-1500 sequencing, and data preprocessing

Pv11 cells were maintained in IPL-41 medium supplemented with 10% FBS (T0) and were incubated in 600 mM trehalose solution containing 10% of supplemented IPL-41 medium for 12, 24, 36, or 48 h (T12, T24, T36, T48). The cells were desiccated in a desiccator for 10 days (R0; relative humidity, <10%). After desiccation, they were rehydrated with IPL-41 containing 10% FBS for 3, 12, 24, or 72 h (R3, R12, R24, R72). Samples were prepared in biological triplicate. Total RNA was extracted with a TruSeq RNA sample prep kit v2 (Illumina, San Diego, CA, USA) and purified with a NucleoSpin RNA kit (Macherey-Nagel, Düren, Germany). RNA quality was assessed with a Bioanalyzer (Agilent Technologies, Santa Clara, CA, USA), and RNA was quantified with a Qubit 2.0 fluorometer (Thermo Fisher, Waltham, MA, USA).

First-strand cDNAs were synthesized and sequenced on an Illumina HiSeq 1500 with a length of 51 nucleotides using a single-end sequencing protocol in the Rapid mode at the Functional Genomics Facility of the National Institute for Basic Biology (Okazaki, Japan). Read quality was checked with FastQC [31]. Based on the results of quality check, the reads were preprocessed by cutting the adaptor region and the first 20 bases from the 3'-end if these bases represented a poly(A) tail; FaQCs [32] was used for preprocessing. Reads with a total length less than 14 bp and reads with more than 2 'N' bases were removed. The remaining reads were mapped to the scaffold of the *P. vanderplanki* genome (midgeBase: http://150.26.71.110/midgebase/) using HISAT2 [33]. The mapping rate was calculated as the ratio of mapped reads to total remaining reads. The mapped reads were counted using the annotated *P. vanderplanki* genome [19] with the *qCount* function in the QuasR package [34].

## Principal component analysis

The read count data of each sample were transformed to reads per kilobase of exon per million mapped reads (RPKM) and used for PCA with the *prcomp* function in R [35]. The PC1–3 scores were calculated for each sample and plotted using the scatterplot3D package [36] in three-dimensional space.

## Analysis of differentially expressed genes

DEG analysis was performed using raw count data of each sample with the *nbinomLRT* function in the DESeq2 package [37]. In the log-likelihood ratio test, the following hypothesis testing was performed for each gene by comparing the mean expression of regarding gene in the overall time-series of trehalose treatment or rehydration, and the expression of regarding gene in each time point.

$H_0$: The mean expression of a gene is the same in overall time points (Null hypothesis).

$H_1$: The mean expression of a gene at least at a one-time point was different from its mean expression in overall time points (Alternative hypothesis).

DEGs were detected when the false discovery rate of the one-sided test calculated by the BH method [38] was less than 0.05. DEGs were classified as those for transcription factors if annotated with GO:0003700 (sequence-specific DNA binding transcription factor activity) [39], or as others if they were not. The numbers of DEGs for transcription factors and other DEGs were visualized as a Venn diagram using the *venn.diagram* function in the VennDiagram package [40].

## Inference of transcriptional regulatory network based on time-series expression data

The regulatory relationships between transcription factors detected as DEGs were inferred using time-series RPKM of each transcription factor in trehalose pretreatment or rehydration using BTNET, which currently has the best performance [41]. Gradient boosting in BTNET was used. The number of boosting trees was set to 100 and the time lag of BTNET was set to 1. Determination of the threshold of the obtained score for each regulation inferred by BTNET was based on the fact that the transcriptional regulatory networks of *Escherichia coli* and *Saccharomyces cerevisiae* have scale-free topology [42]. The scale-free property of the network was defined as

$$p(k) = Ck^{-\gamma} \tag{1}$$

where $k$ denotes the degree (the number of regulations of the transcription factor), $\gamma$ is the coefficient of scale-free topology, and $C$ is the normalization coefficient. The above equation can be log-log transformed as follows:

$$log_{10}p(k) = -\gamma log_{10}k + log_{10}C \qquad (2)$$

Based on the transformed equation, the scale-free property of the network constructed by adopting the regulation whose score was more than score $a$ was evaluated by linear regression between degree and degree distribution using the *lm* function in R. At this step, $\gamma$, the *p*-value of one-sided *F*-test, and the $R^2$ value were calculated for $a$ from 0 to 1 per estimated score (S5 and S6 Figs). Finally, the minimum score satisfying the condition that $\gamma$ was negative and the *p*-value of the *F*-test was less than 0.05 was determined as the threshold of the estimated score by BTNET. The positive or negative regulations were classified based on the sign of the value of cross-correlation with a time lag of 1 using the *ccf* function in R, i.e., the regulation was considered positive when its value was more than 0 and negative when it was less than 0. The estimated transcriptional regulatory networks for trehalose pretreatment and rehydration were integrated to extract the transcription factors and regulation present at least in one of the estimated networks. The transcriptional regulatory networks were visualized using yEd [43]. FFLs and FBLs were extracted using the *graph.get.subisomorphism.vf2* function in the igraph package [44]. The extracted FFLs were classified into coherent or incoherent when the signs of direct and indirect regulation were the same or opposite, respectively. The extracted FBLs were classified into positive or negative when the number of negative regulations was even or odd, respectively. Whether these numbers were significantly higher than that of a random network was tested using a randomization test as follows. A random network whose number of nodes and edges was exactly the same as the number of transcription factors and regulations in the integrated network was generated on the basis of the Erdös–Renyi model [45] using the *erdos.renyi.game* function in the igraph package. The sign of each regulation in the random network was based on a binomial distribution under the condition that the probability of success in the binomial distribution was equal to the ratio of positive regulations to that of all regulations in the integrated network. The numbers of coherent and incoherent FFLs and positive and negative FBLs in the random network were calculated as for the integrated network. These calculations were iterated 10,000 times and the networks with a higher number of each type of FFL and FBL than that of integrated network were counted (S7 Fig). Finally, the *p*-value was calculated for each FFL and FBL by dividing the counts for this structure by 10,000; *p*-values less than 0.05 were considered significant.

## Detection of modules and gene ontology enrichment analysis

One transcription factor binds regulatory regions of various genes and induces the coordinated expression (coexpression) of these genes, called "gene batteries" [46], which we refer to as modules. To detect such modules, we performed weighted gene coexpression network analysis (WGCNA) [47]. For all DEGs other than those for transcription factors, a pair-wise correlation matrix was computed using time-series RPKM of each gene in trehalose pretreatment and rehydration. Using the *pickSoftThreshold* function in the WGCNA package, we chose the *power* of 7 to satisfy the scale-free topology criterion so that the signed $R^2$ value of the fitting of the scale-free equation was more than 0.75 (S8 Fig). Then the adjacency matrix for each pair of genes was obtained by raising the correlation matrix to the *power* of 7. This adjacency matrix was transformed into a robust measure of network interconnectedness (topological overlap measure, TOM [48]). The dissimilarity of each pair of genes was calculated by subtracting 1

from each element of the TOM-based adjacency matrix. This dissimilarity matrix was used as an input for average linkage hierarchical clustering using *hclust* in R (S9 Fig).

Modules were detected with the *cutreeDynamic* function in the WGCNA package. This function requires setting the minimum number of genes per module, minClusterSize. We determined this value as the number that increased the pseudo-F [49], a clustering metric, from 1 to 200; the minClusterSize value based on this criterion was 165 (S10 Fig). Finally, 27 clusters were detected as modules. Each module was assigned an easily recognizable color. To clarify what functional genes belong to each module, we performed GO enrichment analysis using the *GSEAGOHyperGParams* function in the GOstats package [50], and GOs with *p*-values < 0.05 were detected as enriched in each module (S3 Data).

## Prediction of the regulation of coexpressed genes in a module by a transcription factor

Transcription factor–binding motif enrichment analysis was performed with CLOVER [25]. The information on binding motif for each transcription factor of *P. vanderplanki* is not yet available; therefore, we collected information for *D. melanogaster* for each transcription factor from the CISBP [51], FlyFactorSurvey [52], and JASPER2018 [53] databases using motifDB package [54]; we collected 134, 291, and 140 binding motifs, respectively. Then we extracted amino acid sequences of the corresponding transcription factors and searched for sequence similarities against the whole genome of *P. vanderplanki* using bidirectional search in blastp (*e*-value < 1.0e-5). We collected 72 transcription factors of *P. vanderplanki* homologous to *D. melanogaster* proteins, which were linked to the binding motif information. Using these binding motifs and the upstream regions of genes in each module as an input to CLOVER, we performed motif enrichment analysis. The upstream regions were designated as 100, 500, 1,000, 2,000, 3,000, 4,000, 5,000, and 10,000 bp, and the scaffold of *P. vanderplanki* was used as a control sequence. A transcription factor was considered to bind when the *p*-value of one-sided test in CLOVER was less than 0.05 for at least one upstream region, but whether this binding induces gene transcription in the module was not tested. To take this into consideration, we tested Granger causality [55] between the transcription factor and the genes in the module in the time-series expression data for trehalose pretreatment or rehydration. In this test, we transformed all RPKM values for each DEG into Z-score and created a vector autoregressive (VAR) model, in which the time-series Z-score of a transcription factor was set as the cause and the mean time-series Z-score of genes in the module was set as the effect. The VAR model was created with the *VAR* function in the vars package [56], and the time lag of the model was optimized with the *VARSelect* function in the same package for each candidate transcription factor–module pair. We tested whether the variance from the above model was significantly smaller than that of the null model using one-sided *F*-test with the *causality* function in the vars package. The Granger causality was detected when the adjusted *p*-value of the *F*-test by the BH method was less than 0.05. The regulatory relationship was detected if it passed both tests, motif enrichment analysis and Granger causality test.

## Comparative analysis of transcriptional regulatory networks between Pv11 cells and *D. melanogaster* and contraction of the estimated network

To reveal the part of the regulatory network specific to desiccation tolerance, we compared the transcriptional regulatory networks (nodes and edges) between Pv11 cells and *D. melanogaster*. That of *D. melanogaster* was obtained from modEncode (http://intermine.modencode.org/release-33/flyRegulatoryNetwork.do, network was shown in Fig 7A in [26]). To compare the nodes, we examined sequence similarity between *P. vanderplanki* and *D. melanogaster*

transcription factors included in the modEncode network (blastp, *e*-value < 1.0e-15). To compare the edges, the regulatory relationships were judged to exist in *P. vanderplanki* when either direct or indirect regulation from the source to target transcription factors of *D. melanogaster* homologous to those of *P. vanderplanki* existed in the modEncode network. To extract the structure important for desiccation tolerance, we contracted the obtained network. We extracted the transcription factors that had no homology with the transcription factors in modEncode and that directly regulated the modules, and selected those that satisfied any of the following criteria: 1) the most upstream transcription factors, i.e., those with no indegree edge, 2) FFL components, and 3) HSF and transcription factors to connect the transcription factors extracted based on criteria 1) and 2). We selected the regulations that were judged to exist only in *P. vanderplanki*. Finally, we presented each FFL in the contracted network as one block and eliminated the transcription factors between HSF and FFLs.

## Supporting information

**S1 Table. Results of mapping for each sample.**
(XLSX)

**S2 Table. Modules with the maximum number of anhydrobiosis-related genes.**
(XLSX)

**S1 Data. DEGs in trehalose pretreatment and rehydration.**
(XLSX)

**S2 Data. List of *P. vanderplanki* genes in each module.**
(XLSX)

**S3 Data. GO terms enriched in each module.**
(XLSX)

**S4 Data. BLASTN results for the known anhydrobiosis-related genes annotated in a previous study [14] vs. genes of *P. vanderplanki* annotated in a previous study [19], and the module for each gene.**
(XLSX)

**S5 Data. Transcription factor–binding elements in *P. vanderplanki* genes included in modules detected by motif enrichment analysis (CLOVER) and Granger causality test.**
(XLSX)

**S1 Text. This document contains detailed information about the results of GO enrichment analysis for the module that includes anhydrobiosis-related genes reported previously.**
(PDF)

**S1 Fig. Mapping rate to the chromosomal and mitochondrial genomes of *P. vanderplanki* for each sample.** Three biological replicates per sample are shown. The mapping rate to the mitochondrial genome increased sharply just after rehydration.
(PDF)

**S2 Fig. Transcriptional regulatory network inferred for trehalose pretreatment.** Each rectangular node refers to a transcription factor; transcription factor IDs are as defined in [19]; arrows show the inferred regulatory relationships (red, positive; blue, negative). The node color was based on the value of the mean Z-score in each sample calculated at each time point as reads per kilobase of exon per million mapped reads (RPKM) for each gene: red, high

Z-score; blue, low Z-score.
(PDF)

**S3 Fig. Transcriptional regulatory network for rehydration.** Designations are as in S2 Fig.
(PDF)

**S4 Fig. Comparison of the transcriptional regulatory networks between *D. melanogaster* and Pv11 cells.** Rectangular nodes, transcription factors; circular nodes, modules. Transcription factors with sequence similarity to those in the *D. melanogaster* transcriptional regulatory network are shown in green (blastp, *e*-value < 1.0e-15); other transcription factors are shown in red. Regulatory relationships confirmed in both *P. vanderplanki* and *D. melanogaster* are shown as green arrows, and those specific for *P. vanderplanki* are shown as red arrows. Regulatory relationships between transcription factors and modules are shown as grey arrows (CLOVER, *p*-value < 0.05 and Granger causality test, adjusted *p*-value < 0.05, Benjamini-Hochberg method). Thus, the candidate regulatory network specific to desiccation tolerance is represented by red nodes and arrows.
(PDF)

**S5 Fig. Determination of the threshold for the score of inferred regulation by BTNET in trehalose pretreatment.** (upper left) The value of $\gamma$, which is the property of scale-free topology. (upper right) *p*-value for $\gamma$ in *F*-test. (lower left) $R^2$ values obtained by fitting linear relationships between degree (the number of regulations of the transcription factor) and degree distribution. Red lines show the minimum value of the threshold (0.160963) at which $\gamma$ was negative and *p*-value was less than 0.05. (lower right) A plot of the log-log transformed degree distribution based on threshold = 0.160963. The line shows fitting result for the scale-free equation.
(PDF)

**S6 Fig. Determination of the threshold for the score of inferred regulation by BTNET in rehydration.** For panel descriptions and designations, see S5 Fig. The threshold value was 0.151133.
(PDF)

**S7 Fig. Results of the randomization test of the significant number of FFLs and FBLs in the inferred transcriptional regulatory network obtained by integration of those of trehalose pretreatment and rehydration.** Histograms show the number of networks generated by the Erdös–Renyi model (#) categorized against the number of specific structures in the network. Red columns show the numbers of generated networks obtaining more than that of the inferred transcriptional regulatory network obtained by integration of those of trehalose pretreatment and rehydration. FFL, feed-forward loop; FBL, feedback loop. Rates of the red columns: coherent FFLs, 0.0177; incoherent FFLs, 0.5331; positive FBLs, 0.3087; negative FBLs, 0.3009. Thus, only the number of coherent FFLs was significant in the integrated network at the significance level of $\alpha = 0.05$.
(PDF)

**S8 Fig. Signed $R^2$ value calculated by fitting the scale-free equation against the *power* value in weighted gene coexpression network analysis (WGCNA).** 7 of *power* satisfied that the signed $R^2$ was more than 0.75 (red line).
(PDF)

**S9 Fig. Dendrogram of dissimilarity matrix calculated by topological overlap measurements (TOM) subtracted by 1 in WGCNA analysis.** TOM was calculated from the Pearson

correlation matrix of time-series Z-score for gene pairs raised by the estimated *power* of 7.
(PDF)

**S10 Fig. Change of the clustering metric, pseudo-F, for each minModuleSize of a dynamic tree cut from the dendrogram.** Pseudo-F increased from minModuleSize 1 to 165 and drastically decreased after 166; therefore, the minModuleSize was determined as 165.
(PDF)

## Acknowledgments

We are grateful to Pavel Mazin for providing a detailed dataset of the annotated *P vanderplanki* genome, to Jun Okada and Yuki Kikuzato-Sato for preparing RNA-seq libraries, to Tomoe Shiratori for maintaining Pv11 cells, and to Dr. Tomoko Shibata, Ms. Shoko Ohi, Dr. Shuji Shigenobu, Dr. Katsushi Yamaguchi, and Dr. Asaka Akita (NIBB) for sequencing libraries.

## Author Contributions

**Conceptualization:** Takahiro Kikawada, Akira Funahashi.

**Data curation:** Takahiro G. Yamada, Takahiro Kikawada.

**Formal analysis:** Takahiro G. Yamada.

**Funding acquisition:** Takahiro Kikawada, Akira Funahashi.

**Investigation:** Takahiro G. Yamada, Takahiro Kikawada, Akira Funahashi.

**Methodology:** Takahiro G. Yamada.

**Project administration:** Takahiro Kikawada, Akira Funahashi.

**Resources:** Takahiro Kikawada, Akira Funahashi.

**Software:** Takahiro G. Yamada, Yusuke Hiki.

**Supervision:** Takahiro Kikawada, Akira Funahashi.

**Validation:** Takahiro G. Yamada, Takahiro Kikawada.

**Visualization:** Takahiro G. Yamada.

**Writing – original draft:** Takahiro G. Yamada.

**Writing – review & editing:** Yusuke Hiki, Noriko F. Hiroi, Elena Shagimardanova, Oleg Gusev, Richard Cornette, Takahiro Kikawada, Akira Funahashi.

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
