## [Decision Letter · Decision Letter 0]

9 Jan 2020

PONE-D-19-27695

Identification of a master transcription factor and a regulatory mechanism for desiccation tolerance in the anhydrobiotic cell line Pv11

PLOS ONE

Dear Dr. Funahashi,

Thank you for submitting your manuscript to PLOS ONE. After careful consideration, we feel that it has merit but does not fully meet PLOS ONE’s publication criteria as it currently stands. Therefore, we invite you to submit a revised version of the manuscript that addresses the points raised during the review process.

The manuscript has been evaluated by two reviewers, and their comments are available below. The reviewers have raised a few issues that they feel a revised submission should address. In particular, they feel that there are some parts of the data analyses that could be clarified. Please revise the manuscript to carefully address the concerns raised.

We would appreciate receiving your revised manuscript by Feb 23 2020 11:59PM. To enhance the reproducibility of your results, we recommend that if applicable you deposit your laboratory protocols in protocols.io, where a protocol can be assigned its own identifier (DOI) such that it can be cited independently in the future. For instructions see: http://journals.plos.org/plosone/s/submission-guidelines#loc-laboratory-protocols

We look forward to receiving your revised manuscript.

Kind regards,

Hanna Landenmark

Associate Editor

PLOS ONE

Journal Requirements:

2. Please provide details of the origin and vendor information for the cell line used.

Reviewers' comments:

Reviewer's Responses to Questions

**Comments to the Author**

1. Is the manuscript technically sound, and do the data support the conclusions?

Reviewer #1: Yes

Reviewer #2: Yes

2. Has the statistical analysis been performed appropriately and rigorously? 

Reviewer #1: Yes

Reviewer #2: Yes

3. Have the authors made all data underlying the findings in their manuscript fully available?

Reviewer #1: Yes

Reviewer #2: Yes

4. Is the manuscript presented in an intelligible fashion and written in standard English?

Reviewer #1: Yes

Reviewer #2: Yes

5. Review Comments to the Author

Reviewer #1: The anhydrobiosis is an outstanding phenomenon in animal and plant. Nevertheless, many efforts were invested to analyze the genetic basis of desiccation tolerance, transcriptional regulatory mechanisms underlying induction of desiccation-associated genes are still poorly studied. The authors thoroughly analyzed gene regulatory network in embryonic cell line Pv11 of African midge, Polypedilum vanderplanki, and compared the obtained results with that of Drosophila melanogaster. The main goals are clearly enunciated. Research design and methodologies in addressing the raised questions are very well thought out, appropriate and fitting. A huge amount of strong and convincing data and deep and wide discussion of desiccation tolerance mechanisms are presented in the paper. The experimental procedures are described so well and scrupulous that no questions remain unanswered. The style of writing makes it easily understood by a diverse audience of scientists. Data presentation looks attractive, elegant and convincing. Figures and tables are clear and sufficiently illustrative. The suggested approach and obtained results will certainly forward understanding of desiccation tolerance mechanisms.

I strongly support publication.

Reviewer #2: This study uses a cell line from a highly desiccation tolerant insect to investigate gene regulatory networks underlying anhydrobiosis. Nuclear transcription factor Y subunit gamma-like (NF-YC) is identified as a key upstream regulator of feedback loops regulating anhydrobiosis-related genes. NF-YC is known to be important in plant drought tolerance; this study is an important addition to the list of common mechanisms (e.g. LEA proteins, antioxidant processes) responsible for dehydration-tolerance in plants and animals.

As a physiologist, I must state at the start that I am not very familiar with systems biology and have only recently been collaborating with a systems biologist. I outlined the general analysis trajectory of this study to her, and she felt (without knowing details) that the approach seemed reasonable. I will note, except for the section on Inference of transcriptional regulatory network…… (page 16/28), I found the Materials and methods to follow a clear logical sequence.

Experimental methods for RNA isolation and sequencing were standard. The differential expression analysis was a bit unclear. Fold-change differences are provided for each treatment. Differences compared to what? Untreated cells? Are these the data used to generate S2_Fig and S3_Fig?

The GO enrichment analysis (S1_Text) was restricted to genes previously identified as being involved in anhydrobiosis. This seems to be biased to confirm previous work. There are many unbiased GO enrichment methods published – why not look for novel GO categories that could lead to new understanding?

I like the comparison to the Drosophila melanogaster transcriptional regulatory network, but I was left wondering: which D. melanogaster network? Citation 26 includes one in the full text and 4 in the supplemental material.

The Discussion was concise and clearly placed the results of this work in the greater context of anhydrobiosis. In general, I found the entire manuscript to be well written and well edited.

6. PLOS authors have the option to publish the peer review history of their article (what does this mean?). If published, this will include your full peer review and any attached files.

Reviewer #1: No

Reviewer #2: No

---

## [Author Response · Author response to Decision Letter 0]

15 Jan 2020

Reply to Reviewer #1

Thank you so much for your positive comments.

Reply to Reviewer #2

Thank you for your precious comments. We hope that our revised manuscript will fulfill your request.

Minor comments

1. Experimental methods for RNA isolation and sequencing were standard. The differential expression analysis was a bit unclear. Fold-change differences are provided for each treatment. Differences compared to what? Untreated cells? Are these the data used to generate S2_Fig and S3_Fig?

Response: 

You are correct that the explanation about DEG analysis was unclear in terms of the sample comparison. In this study, the detection of DEGs was performed by comparing the mean expression of a gene in the overall time-series of trehalose treatment or rehydration, and its expression in each time point. That means a gene is detected as DEG when the mean expression of the gene at least at a one-time point was different from the mean expression in overall time points. Therefore, we did not compare the expression of a gene with that from untreated cells. In S2_Fig and S3_Fig, there are nodes (transcription factors) that were detected as DEG based on the above DEG analysis. Clarifying these points, we have added the sentence to our manuscript in materials and methods "Analysis of differentially expressed genes'' (pp. 16, line 264 to 271). 

2. The GO enrichment analysis (S1_Text) was restricted to genes previously identified as being involved in anhydrobiosis. This seems to be biased to confirm previous work. There are many unbiased GO enrichment methods published – why not look for novel GO categories that could lead to new understanding?

Response:

Firstly, we declare that the result of GO enrichment analysis for each module was not used for any prior knowledge to extract the finally contracted gene regulatory network (Fig. 8). This result was used only for the verification of whether GOs related to the functions previously revealed as important for desiccation tolerance are enriched in modules or not (S1 Text). As you pointed out, there are various GOs that were not focused on this verification (S3 Data). The reason why we did not analyze it in detail is that we mainly focused on finding not the mechanism regulated by effector genes but the transcriptional regulatory mechanism in this study. However, your point is undoubtedly right in terms of uncovering the mechanism regulated by effector genes. Therefore, we have added the sentence that "The subsequent detailed analysis of these GOs could lead to the discovery of novel molecular biological mechanisms of desiccation tolerance. '' and aforementioned scope of this research in Discussion (pp. 12, line 167 to 176).

3. I like the comparison to the Drosophila melanogaster transcriptional regulatory network, but I was left wondering: which D. melanogaster network? Citation 26 includes one in the full text and 4 in the supplemental material.

Response:

Your point is correct. The explanation of the resource about the transcriptional regulatory network of D.melanogaster was not enough. We used Fig. 7A in the full text of citation 26. We downloaded it from modEncode database, and the URL is http://intermine.modencode.org/release-33/flyRegulatoryNetwork.do. The main reason to use it was that this network was the most comprehensive transcriptional regulatory network compared with four networks in supplementary materials of citation 26. We have added this information in materials and methods "Comparative analysis of transcriptional regulatory networks between Pv11 cells and D. melanogaster and contraction of the estimated network'' (pp. 20, line 386 to 387).

---

## [Decision Letter · Decision Letter 1]

25 Feb 2020

Identification of a master transcription factor and a regulatory mechanism for desiccation tolerance in the anhydrobiotic cell line Pv11

PONE-D-19-27695R1

Dear Dr. Funahashi,

We are pleased to inform you that your manuscript has been judged scientifically suitable for publication and will be formally accepted for publication once it complies with all outstanding technical requirements.

With kind regards,

Axel Imhof

Academic Editor

PLOS ONE

Additional Editor Comments (optional):

Reviewers' comments:

Reviewer's Responses to Questions

**Comments to the Author**

1. If the authors have adequately addressed your comments raised in a previous round of review and you feel that this manuscript is now acceptable for publication, you may indicate that here to bypass the “Comments to the Author” section, enter your conflict of interest statement in the “Confidential to Editor” section, and submit your "Accept" recommendation.

Reviewer #1: All comments have been addressed

Reviewer #2: All comments have been addressed

2. Is the manuscript technically sound, and do the data support the conclusions?

Reviewer #1: Yes

Reviewer #2: Yes

3. Has the statistical analysis been performed appropriately and rigorously? 

Reviewer #1: Yes

Reviewer #2: Yes

4. Have the authors made all data underlying the findings in their manuscript fully available?

Reviewer #1: Yes

Reviewer #2: Yes

5. Is the manuscript presented in an intelligible fashion and written in standard English?

Reviewer #1: Yes

Reviewer #2: Yes

6. Review Comments to the Author

Reviewer #1: (No Response)

Reviewer #2: (No Response)

7. PLOS authors have the option to publish the peer review history of their article (what does this mean?). If published, this will include your full peer review and any attached files.

Reviewer #1: No

Reviewer #2: No

---

## [Editor Report · Acceptance letter]

4 Mar 2020

PONE-D-19-27695R1 

Identification of a master transcription factor and a regulatory mechanism for desiccation tolerance in the anhydrobiotic cell line Pv11 

Dear Dr. Funahashi:

I am pleased to inform you that your manuscript has been deemed suitable for publication in PLOS ONE. Congratulations! Your manuscript is now with our production department. 

With kind regards,

on behalf of

Dr. Axel Imhof 

Academic Editor

PLOS ONE